# Neurological events and unanticipated risks after locoregional anesthesia (NEURAL): Protocol for a multicenter prospective observational study

**Alessandro De Cassai**[1,2*], **Elena Ioppolo**[3], **Dario Bugada**[4], **Francesco Tasso**[5], **Gianluca Cappelleri**[6], **Vito Torrano**[7]

1 Department of Medicine (DIMED), University of Padua, Padua, Italy, 2 Institute of Anesthesia and Intensive Care, University Hospital of Padua, Padua, Italy, 3 School of Medicine and Surgery, University of Milano-Bicocca, Milan, Italy, 4 Department of Emergency and Critical Care, ASST Papa Giovanni XXIII, Bergamo, Italy, 5 IRCCS Humanitas Research Hospital, Rozzano, Italy, 6 Policlinico di Monza Hospital, Monza, Italy, 7 Department of Anesthesia, Critical Care and Pain Medicine, ASST Grande Ospedale Metropolitano Niguarda, Milan, Italy

* alessandro.decassai@unipd.it

## Abstract

### Background

Regional anesthesia is widely regarded as one of the safest anesthetic techniques, yet serious complications—such as nerve injury, hematoma, pneumothorax, and local anesthetic systemic toxicity (LAST)—continue to be reported. Their true incidence remains uncertain, as available data are often derived from registry or retrospective studies with heterogeneous definitions and limited sample sizes. Moreover, the mechanisms underlying complications such as nerve injury are incompletely understood and may extend beyond direct mechanical trauma to include factors such as sub-perineural injection, hematoma formation, altered coagulation, and patient-specific vulnerability.

### Methods

The *NEURAL* study, promoted by the Italian Society of Anesthesia, Analgesia and Critical Care (SIAARTI), is a multicenter, prospective, observational study designed to determine the incidence and risk factors for complications following single-shot regional anesthesia of the upper limb, lower limb, and fascial plane. The primary endpoint is the composite incidence of nerve injury, hematoma, pneumothorax, and LAST. Secondary objectives include determining the individual incidence of each complication and identifying patient- and procedure-related risk factors. Data will be collected via the REDCap® (Research Electronic Data Capture) platform from more than 40 Italian centers. Standardized follow-up will be performed at 24 and 48 hours, 15 and 30 days, and monthly thereafter for unresolved neurological deficits, up to one

**Data availability statement:** No datasets were generated or analysed during the current study. All relevant data from this study will be made available upon study completion.

**Funding:** The author(s) received no specific funding for this work.

**Competing interests:** The authors have declared that no competing interests exist.

year. Statistical analyses will include logistic regression modeling to identify independent predictors of complications.

## Expected results

Based on an estimated complication rate of 0.5%, a minimum of 3,396 patients will be required to ensure adequate precision of incidence estimates. Nationwide participation is expected to exceed this target.

## Conclusions

The *NEURAL* study will provide robust, prospective, and standardized data on complications of regional anesthesia. By identifying their true incidence and modifiable risk factors, the findings are expected to inform safer clinical practice, enhance patient counseling, and support the development of updated evidence-based guidelines.

## Introduction

Regional anesthesia has an extremely high safety profile [1]. However, complications related to regional techniques are reported in the literature such as local anesthetic toxicity, hematoma requiring medical attention or nerve injury [2–4]. Given that such complications are extremely rare their incidence is not fully known, mainly because it is estimated based on registry studies with an often limited sample size.

Moreover, while the cause-effect mechanism of regional anesthesia is clear for some complications – for instance, a pneumothorax following a paravertebral block –, the mechanism of action for other complications, such as nerve injury after regional anesthesia, is not well understood. Historically, it was believed that nerve injury resulted from the needle coming into direct contact with the nerve fibers, essentially causing mechanical trauma. Nevertheless, recent evidence challenges this view, showing that it is mechanically difficult to cause nerve injury solely through direct contact. In fact, nerve damage can occur even without any direct interaction with the nerve [5].

Thus, in addition to factors like local anesthetic toxicity, sub-perineural injection, high injection pressures within the epineurium, and sub-epineural hematoma resulting from forced contact between the needle and the nerve, other contributing factors may also play a role in nerve damage. These include altered coagulation profiles, the patient's intrinsic fragility, possible comorbidities, and other currently unidentified factors [6].

The aim of the NEURAL study will be to provide a comprehensive estimate of the incidence of complications following regional anesthesia, with particular attention to nerve injury, hematoma, pneumothorax, and local anesthetic systemic toxicity (LAST). Moreover, by conducting a large multicenter prospective observational study, we will also seek to identify patient- and procedure-related risk factors that may predispose individuals to neurological or systemic adverse events. We hypothesize that

the true incidence of complications is higher than currently appreciated from registry data, and that specific risk factors—including comorbidities, anticoagulant use, and technical variables related to block performance—contribute significantly to their occurrence.

## Materials and methods

We are reporting the study protocol for the NEURAL study. The study had not yet started at the time of this protocol's submission to the journal. It is tentatively scheduled to begin in January 2026 and to run for two consecutive years (until January 2028). Study results are expected to be definitive at the end of 2028. The final start date will be updated on ClinicalTrials.gov as soon as it is confirmed.

### Study design and objectives

This protocol has been prepared with the support of the SPIRIT checklist (see S1 File). The NEURAL study, promoted by the Italian Society of Anesthesia, Analgesia and Critical Care (SIAARTI), is a multicenter, prospective observational study designed to investigate the prevalence of complications associated with regional anesthesia of the upper limb, lower limb, and fascial plane blocks.

The primary objective of the NEURAL study is to determine the overall incidence of complications following regional anesthesia, expressed as a composite outcome. This composite outcome will include nerve injury, hematoma, pneumothorax, and LAST, calculated as the total number of complication events divided by the total number of procedures.

The secondary objectives are to determine the incidence of each individual complication (nerve injury, hematoma, pneumothorax, and LAST for each type of regional anesthesia technique. These include upper limb blocks, lower limb blocks, and fascial plane blocks (Table 1), with outcomes calculated as the number of events relative to the total number of procedures performed. In addition, the study seeks to identify risk factors associated with complications following upper limb regional anesthesia, lower limb regional anesthesia, and fascial plane blocks, thereby contributing to a more precise understanding of patient- and procedure-related variables influencing safety in regional anesthesia as described in the relevant paragraph below..

The regional anesthesia techniques will be categorized using the most recent nomenclature proposed by the European Society of Regional Anesthesia (ESRA) [6–7] as shown in Table 1

### Definitions

Complications will be defined as follows:

- Hematoma: A blood collection at the regional anesthesia site requiring medical and/or surgical intervention.

- Pneumothorax: The presence of an air crescent consistent with pneumothorax on the same side as the regional anesthesia, confirmed by ultrasound, chest X-ray, or chest CT scan.

- LAST: Any electrocardiographic, hemodynamic, and/or neurological alteration occurring after regional anesthesia, attributable to systemic absorption of local anesthetic and requiring treatment (e.g., lipid emulsion administration).

- Nerve injury: The onset of dysesthesia, anesthesia, or prolonged motor deficit not explained by the pharmacokinetics of the anesthetic used, within the distribution area consistent with the performed block.

### Inclusion and exclusion criteria

All adult patients (≥18 years) undergoing single-shot regional anesthesia of the upper limb, lower limb, or fascial plane will be considered eligible for inclusion after providing informed consent.

**Table 1. Fascial plane regional anesthesia: The most recent nomenclature of the European Society of Regional Anaesthesia (ESRA) [7] will be used. The following blocks will be considered.**

| Upper limb blocks | Lower limb blocks | Fascial blocks |
|---|---|---|
| Interscalene BP block | Lumbar plexus block | Rectus sheath block |
| Superior trunk block Supraclavicular BP block InfraclavicularBP block | Sacral plexus block | Ilioinguinal iliohypogastric nerves block |
| Infraclavicular BP block (coracoid approach) | Fascia iliaca block (suprainguinal approach) | TA block |
| Infraclavicular BP block (retroclavicular approach) | Fascia iliaca block (infrainguinal approach) | Midaxillary TAP block |
| Infraclavicular BP block (costoclavicular approach) | Adductor canal block | Subcostal transversus abdominis plane block |
| Suprascapular nerve block (anterior approach) | PENG block | ESP block |
| Suprascapular nerve block (posterior approach) | Femoral nerve block | Deep SAP block |
| Axillary BP block | Femoral triangle block | Superficial SAP block |
| Superficial cervical plexus block | Sciatic nerve block (anterior approach) | TFP block |
| Intermediate cervical plexus block | Sciatic nerve block (transgluteal approach) | Rhomboid intercostal plane block |
| Deep cervical plexus block. | Sciatic nerve block (infragluteal approach) | Retrolaminar block |
| | Sciatic nerve block at the popliteal fossa | Anterior QLB |
| | Nerve to vastus medialis block | Lateral QLB |
| | Genicular nerves block | Posterior QLB |
| | IPACK | Paravertebral block |
| | Common peroneal nerve block | ITP block |
| | Ankle block | Superficial PIP block |
| | Pudendal nerve block. | DeepPIP block; |
| | | PSP block |
| | | IPP block |

BP:brachial plexus; ESP: Erector spinae plane; IPACK: Infiltration between the popliteal artery and capsule of the knee; IPP: Interpectoral plane;ITP: Intertransverse process; PENG: Pericapsular Nerve Group; PIP: parasternal intercostal plane; PSP: Pectoserratus plane; QLB: quadratus lumborum block; SAP: serratus anterior plane; TAP:Transversus abdominis plane; TFP: Transversalis fascia plane.

Patients will be excluded if they are not able to provide informed consent or if they are receiving multiple regional anesthesia techniques. However patients receiving more than one single-shot regional block during the same operative session may be included if the blocks do not share potential complications or overlapping sensory innervation. For example, a patient undergoing both an interscalene plexus block for shoulder surgery and a femoral nerve block for knee surgery may be included, as the two blocks involve distinct innervation areas and complication profiles. Conversely, patients receiving both a paravertebral block and an erector spinae plane (ESP) block on the same side of the chest would not be eligible, since these techniques share common risks (e.g., pneumothorax) and overlapping innervation, making it impossible to attribute complications to a specific block. However, a patient receiving a paravertebral block on one side of the chest and an ESP block on the opposite side could be included, as no overlap in innervation or complication risk would be expected in this scenario.

## Recruitment

Participants will be recruited from as many centers as possible, subject to approval by their respective ethics committees. The centers initially involved in the steering committee include: Azienda Ospedale–Università di Padova, Padua, Italy; ASST Grande Ospedale Metropolitano Niguarda, Milan, Italy; ASST Papa Giovanni XXIII, Bergamo, Italy; and Humanitas Rozzano, Rozzano (MI), Italy. At the time of submission of this protocol, 43 centers had joined the study. The list of centers participating in the NEURAL study will be continuously updated on the official *ClinicalTrials.gov* record. (the complete list of participating centers is available on *ClinicalTrials.gov*). The list of centers participating in the NEURAL study will be continuously updated on the official *ClinicalTrials.gov* record (NCT07238933; first submitted on 02.10.2025, publicly available since 20.11.2025) as new centers join the study.

## Data collection

Following informed consent, data will be collected using a standardized collection form. Each data entry will be generated by a participating center and recorded on SIAARTI servers through the REDCap® (Research Electronic Data Capture) data management software. REDCap is a secure web-based platform designed to facilitate data capture and management for research studies.

The following data will be registered:

## Intraoperative time

- Demographics: age (years), sex (male/female), BMI (kg/cm$^2$).

- Medical history: relevant comorbidities (i.e., arterial hypertension, diabetes mellitus, peripheral neuropathy, central neuropathy).

- Medication history: with particular attention to anticoagulant and antiplatelet agents, including type and dosage.

- Intraoperative data:

  • pre-procedural sedation (i.e., type of drug), and, if sedation was performed, the level of sedation at the time of block execution — assessed using the Richmond Agitation Sedation Scale (RASS) scale

  • type of regional anesthesia

  • Needle used (sharp vs not-sharp and diameter in gauge)

  • use of ultrasound (yes/no, visualization of structures yes/no, in-plane or out-of-plane approach, nerve swelling on injection yes/no)

  • use of a nerve stimulation device (maximum and minimum intensity used)

  • use of a pressure limiter

  • type of local anesthetic used (volume and dose in mg)

  • single local anesthetic drug or mixture of local anesthetic drugs

  • use of adjuvants (type and dosage in mg or mcg)

  • pain on injection (yes/no)

  • presence (yes/no) of complications, such as hematoma (yes/no) and nerve deficits (yes/no), and their characteristics (sensory – dysesthesia, anesthesia, paresthesia –, or motor deficits in the affected area) at 24 and 48 hours.

As nerve deficits may not be immediately apparent and can be masked by factors such as local edema or limb immobilization [7]. Therefore, sensory and motor deficits will be investigated at 15 and 30 days through a standardized telephone questionnaire aimed at assessing sensory loss, motor impairment, and neuropathic pain (see S2 File). Patients who report the presence of sensory or motor neurological deficits will be referred for management according to the practices of the participating center. For research purposes, they will receive a telephone call every 30 days to monitor the evolution of the deficit until resolution, with a maximum follow-up period of one year.

In summary, the evaluation of complications will require patient contact (either through in-person visits or telephone contact if discharged) at 24 and 48 hours, as well as 15 and 30 days. In the event of nerve complications, the treatment

of the lesion will follow the hospital's local protocol. For the study, follow-up will continue every 30 days until the deficit resolves, or for a maximum of one year from its onset. A timeline is available as Fig 1.

## Statistical analysis

Data for each continuous variable will be analyzed to assess the normality of distribution using the Shapiro-Wilk test. Continuous variables with normal distributions will be reported as mean (standard deviation); those with non-normal distributions will be reported as median and interquartile range. Analysis of normally and non-normally distributed data will be performed using the two-tailed Student's t-test and the Mann-Whitney U test, respectively.

If the proportion of missing data exceeds 5% for key variables, multiple imputation by chained equations will be performed under the assumption of missing at random. Imputation models will include all variables used in the primary analyses, including outcomes and relevant predictors. Sensitivity analyses will be conducted to compare complete-case analyses with imputed datasets. In addition, if differential loss to follow-up is observed, inverse probability weighting methods may be applied to assess the robustness of the findings. The amount of missing data and the methods used to address it will be transparently reported according to STROBE recommendations.

Results for categorical variables will be reported as numbers (percentages) and compared between groups using the chi-squared test or Fisher's exact test, as appropriate.

| | TRIAL PERIOD | | | | | | |
| | Enrollment | | Post-Enrollment (days) | | | | Close-out (months) |
| TIMEPOINT | $-t_i$ to 0 | 0 | 1 | 2 | 15 | 30 | 12 |
| **ENROLLMENT:** | | | | | | | |
| **Eligibility screen** | X | | | | | | |
| **Informed consent** | X | | | | | | |
| **INTERVENTION** | | | | | | | |
| *Regional Anesthesia* | | X | | | | | |
| **ASSESSMENTS:** | | | | | | | |
| *Hematoma* | | X | | | | | |
| *Pneumothorax* | | | X | X | | | |
| *LAST* | | | X | X | | | |
| *Motor/Sensory deficit* | | | X | X | X | X | X |

LAST: Local Anesthetic Systemic Toxicity

**Fig 1. Participant timeline: Schedule of enrollment, intervention, and assessments.**

To determine the strength and direction of association between two variables, the Bravais-Pearson correlation test will be used for variables with a normal distribution, and the Spearman rank correlation test for variables not meeting normal distribution assumptions.

To determine the relationships between the dependent categorical variable (e.g., complications) and one or more independent categorical variables (i.e., complication predictors), logistic regression will be performed to calculate odds ratios (OR) with 95% confidence intervals (CI). Given the anticipated low incidence of complications, rare-event bias may affect conventional maximum likelihood estimation. Therefore, penalized likelihood approaches—such as Firth's bias-reduced logistic regression—will be used when appropriate to improve the stability and validity of parameter estimates. In cases of extremely sparse data, exact logistic regression methods may also be considered. Model adequacy will be assessed using goodness-of-fit statistics and calibration plots. The presence of multicollinearity will be assessed using variance inflation factors.

Given the exploratory nature of this large observational study and the evaluation of multiple block types, predictors, and secondary outcomes, we recognize the potential risk of false-positive findings due to multiple comparisons. The secondary outcomes and subgroup analyses, effect sizes and 95% confidence intervals will be emphasized rather than sole reliance on p-values. Where appropriate, false discovery rate control using the Benjamini-Hochberg procedure may be applied to secondary analyses to reduce the risk of Type I error inflation. All secondary analyses will be interpreted cautiously and considered hypothesis-generating.

**Accounting for center-level clustering.** Given the multicenter design of the NEURAL study, patients are clustered within participating institutions, and outcomes may be correlated due to shared clinical practices, operator expertise, equipment, or local protocols. To account for this hierarchical structure, multilevel (mixed-effects) regression models will be used for the primary and secondary analyses. Specifically, random-intercept logistic regression models will be applied to account for between-center variability when estimating associations between predictors and complications. The intraclass correlation coefficient (ICC) will be calculated to quantify the degree of clustering. As a sensitivity analysis, generalized estimating equations (GEE) with robust standard errors will also be considered.

All statistical analyses will be performed using R software. P-values < 0.05 will be considered indicative of statistical significance.

## Sample size calculation

This is an observational study aimed at investigating very rare complications, the incidence of which is difficult to estimate. Literature reports an incidence ranging from 0 to 2.8%, depending on the study and cohort considered [8].

The sample size estimation was performed using an approach based on the precision of the proportion estimate, defined as the total width of the confidence interval. For this purpose, the Agresti-Coull Wilson method was applied [9].

Starting from a cumulative incidence of 0.005 (0.5%) [10], with a 95% confidence level and aiming for a confidence interval width of 0.003, a minimum of 3,396 subjects must be enrolled. The estimation was conducted using the {presize} package of R software, version 4.3.3 (2024-02-29 ucrt).

When the study was initially planned, we anticipated enrolling at least 10 centers in the NEURAL study, with an expected average recruitment of approximately 350 patients per center over two years, making the achievement of the planned sample size a realistic goal.

## Study duration

The overall duration of the study will be two years.

To ensure adequate standardization, each center participating in the NEURAL study will recruit patients for a duration of six months following ethics committee approval (therefore, each center may recruit patients up to 180 days from the enrollment of the first patient at that center).

### Ethical approval and protocol registration

On May 15, 2025, the study received a conditional favorable opinion from the *Comitato Etico Territoriale Centro-Est Veneto* (Prot. 38516/2025). Subsequently, on September 18, 2025, an amendment was approved, authorizing the participation of the 46 centers currently involved in the study. A copy of the IRB approved protocol is available as S3 File in both original language (Italian) and English.

The study will be conducted according to the principles of the Declaration of Helsinki and subsequent amendments All patients must provide written informed consent prior to inclusion in the study, moreover,in order to comply with the legal provisions set forth by Good Clinical Practice regulations (Legislative Decree 211/2003) and in accordance with the laws and regulations regarding personal data protection pursuant to Article 13 of EU Regulation 679/2016 (GDPR) [11], each patient will receive information about the study and be asked to provide written informed consent for the processing of personal data.

### Additional risks or benefit for the patient

There are no additional risks or benefits for the patient considering the observational design of the study.

### Publication and authorship policy

The Steering Committee will oversee the statistical analysis and synthesis of the collected data and will draft the manuscript for submission to a peer-reviewed journal. Each center will appoint local coordinators responsible for ensuring all necessary ethical and regulatory approvals are obtained, protocol adherence during the study period, and the integrity and completeness of collected data. To recognize the significant contributions of all participating centers and researchers, an Authors' Group will be established under the name "SIAARTI Study Group." The final publication of the study results will acknowledge the contributions of all members of the SIAARTI Study Group. Collaborators and authors will be duly recognized, and their contributions will be traceable through major scientific databases such as PubMed and Scopus.

## Discussion

### Study significance

Regional anesthesia is considered one of the safest anesthetic techniques, offering significant advantages in perioperative pain management and recovery. However, although major complications such as nerve injury, hematoma, pneumothorax, and LAST are rare, their true incidence remains uncertain [12]. Current estimates largely rely on registry data and retrospective analyses, which often suffer from limited sample sizes, heterogeneous reporting, and inconsistent definitions of adverse events [13].

The NEURAL study is designed to provide a comprehensive and prospective evaluation of these complications, contributing new evidence on their prevalence and associated risk factors. By collecting standardized data from a large multicenter cohort, this study will help refine current knowledge about the real-world safety of regional anesthesia and elucidate the mechanisms underlying neurological and systemic adverse events.

### Strengths and limitations

A key strength of the NEURAL study lies in its multicenter, prospective, and observational design, coordinated under the auspices of SIAARTI. The inclusion of multiple tertiary centers across Italy ensures broad representation of clinical practice, patient demographics, and procedural variability, thereby enhancing the generalizability of results.

The study's standardized definitions of complications and structured follow-up at 24 and 48 hours, as well as at 15 and 30 days—with extended follow-up up to one year for nerve injury—will enable accurate detection of both early and delayed complications. The detailed recording of procedural variables (e.g., ultrasound guidance, injection pressure,

adjuvants used) and patient characteristics (e.g., comorbidities, anticoagulant use) allows for in-depth risk factor analysis using multivariable modeling.

However, the study also has inherent limitations. As an observational protocol, it cannot establish causal relationships between regional anesthesia and complications. Additionally, the identification of rare events may still be constrained by the sample size, despite careful power calculation. Potential variability in operator experience and institutional practices may introduce confounding, though standardized definitions and training across centers are intended to minimize this effect.

### Expected impact on clinical practice

The NEURAL study aims to generate high-quality, real-world data that can directly inform clinical practice. By providing precise estimates of complication incidence and identifying modifiable risk factors, it may lead to more targeted preventive strategies, improved patient counseling, and optimization of regional anesthesia techniques. Furthermore, understanding the multifactorial nature of nerve injury—beyond direct mechanical trauma—could influence educational priorities, procedural guidelines, and device development for safer block performance.

Ultimately, the findings of the NEURAL study are expected to strengthen the evidence base supporting the safety of regional anesthesia, guide individualized risk assessment, and promote continuous quality improvement within anesthesiology practice.

### Supporting information

**S1 File. SPIRIT Checklist.**
(DOCX)

**S2 File. Telephone Questionnaire.**
(DOCX)

**S3 File. Approved Protocol both in English and Original Language (Italian).**
(DOCX)

### Author contributions

**Conceptualization:** Alessandro De Cassai, Dario Bugada, Vito Torrano.

**Investigation:** Alessandro De Cassai, Vito Torrano.

**Methodology:** Alessandro De Cassai, Dario Bugada, Vito Torrano.

**Supervision:** Alessandro De Cassai, Vito Torrano.

**Validation:** Francesco Tasso.

**Visualization:** Francesco Tasso.

**Writing – original draft:** Alessandro De Cassai, Elena Ioppolo, Dario Bugada, Francesco Tasso, Gianluca Cappelleri, Vito Torrano.

**Writing – review & editing:** Alessandro De Cassai, Elena Ioppolo, Dario Bugada, Francesco Tasso, Gianluca Cappelleri, Vito Torrano.

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
