## [Decision Letter · Decision Letter 0]

18 Feb 2026

PONE-D-25-62561Neurological events and unanticipated risks after locoregional anesthesia (NEURAL): protocol for a multicenter prospective observational studyPLOS One

Dear Dr. De Cassai,

Thank you for submitting your manuscript to PLOS ONE. After careful consideration, we feel that it has merit but does not fully meet PLOS ONE’s publication criteria as it currently stands. Therefore, we invite you to submit a revised version of the manuscript that addresses the points raised during the review process.

If applicable, we recommend that you deposit your laboratory protocols in protocols.io to enhance the reproducibility of your results. Protocols.io assigns your protocol its own identifier (DOI) so that it can be cited independently in the future. For instructions see: https://journals.plos.org/plosone/s/submission-guidelines#loc-laboratory-protocols. Additionally, PLOS ONE offers an option for publishing peer-reviewed Lab Protocol articles, which describe protocols hosted on protocols.io. Read more information on sharing protocols at . Additionally, PLOS ONE offers an option for publishing peer-reviewed Lab Protocol articles, which describe protocols hosted on protocols.io. Read more information on sharing protocols at https://plos.org/protocols?utm_medium=editorial-email&utm_source=authorletters&utm_campaign=protocols..

We look forward to receiving your revised manuscript.

Kind regards,

Richa Gupta

Academic Editor

PLOS One

Journal Requirements:

2. Please amend either the abstract on the online submission form (via Edit Submission) or the abstract in the manuscript so that they are identical.

Additional Editor Comments:

EDITORIAL DECISION

This is a well-written and engaging manuscript that addresses an important topic. The study is clearly structured, and the findings are presented in a coherent manner. After careful evaluation of the reviewers’ comments and a thorough assessment of the overall quality, originality, and relevance of the submission, the editorial decision is: MINOR REVISION.

The reviewers have provided constructive suggestions aimed primarily at improving clarity, presentation, and minor methodological or interpretative aspects. Please find attached reviewer’s comments:

REVIEWER 1 – MINOR REVISION

The NEURAL study is a large, multicenter prospective observational project designed to determine the true incidence of complications—such as nerve injury, hematoma, pneumothorax, and LAST—following single-shot regional anesthesia. It will enroll over 3,396 adult patients across more than 40 Italian centers, with follow-up extending up to one year for persistent neurological deficits. The study collects detailed patient, procedural, and follow up data to identify both overall incidence and modifiable risk factors. Ultimately, its findings aim to improve patient safety, refine clinical practices, and support updated evidence based guidelines.

Minor revisions:

1. Lack of methods to account for clustering

The protocol does not address the implications of enrolling patients across more than 40 centers, where outcomes may be correlated within institutions due to shared practices, operator skill, equipment, or workflow. Without incorporating statistical methods that account for clustered data—such as multilevel (mixed effects) models, generalized estimating equations (GEE), or random effects logistic regression—standard errors may be underestimated, leading to inflated Type I error rates. Clarifying how center level variation will be handled is essential.

2. No stated strategy for handling missing data

Given the extended follow up period (up to one year for neurological deficits), loss to follow up and incomplete covariate data are likely. The protocol does not describe any approach for addressing missingness, such as multiple imputation, inverse probability weighting, or sensitivity analyses. A predefined plan is necessary to ensure unbiased estimates and transparent reporting.

3. Absence of multiple comparison considerations

With numerous block types, predictors, and secondary outcomes, the probability of false positive findings increases. Although it is reasonable for exploratory observational studies to avoid overly conservative adjustments, the protocol should acknowledge this issue and clarify whether procedures such as Bonferroni correction, false discovery rate control, or hierarchical testing will be considered in interpreting results.

4. Rare event logistic regression concerns

The expected incidence of complications (~0.5%) presents challenges for conventional logistic regression, which can yield biased or unstable estimates when events are sparse. The protocol does not discuss alternative methods tailored to rare events, such as Firth penalized likelihood or exact logistic regression, which may improve model performance and inferential reliability.

REVIEWER 2 – ACCEPT

Dear Authors!

Seems to be a well planed study. I don't say it is absolute necessary, but worth thinking to incorporate the early signs of local anaesthetic systemic toxictiy (or is it included in neurological alteration?) and whether they needed intervention or not.

Reviewers' comments:

Reviewer's Responses to Questions

**Comments to the Author**

1. Does the manuscript provide a valid rationale for the proposed study, with clearly identified and justified research questions?

Reviewer #1: Yes

Reviewer #2: Yes

2. Is the protocol technically sound and planned in a manner that will lead to a meaningful outcome and allow testing the stated hypotheses?

Reviewer #1: Yes

Reviewer #2: Yes

3. Is the methodology feasible and described in sufficient detail to allow the work to be replicable?

Reviewer #1: Yes

Reviewer #2: Yes

4. Have the authors described where all data underlying the findings will be made available when the study is complete?

The PLOS Data policy requires authors to make all data underlying the findings described in their manuscript fully available without restriction, with rare exception, at the time of publication. The data should be provided as part of the manuscript or its supporting information, or deposited to a public repository. For example, in addition to summary statistics, the data points behind means, medians and variance measures should be available. If there are restrictions on publicly sharing data—e.g. participant privacy or use of data from a third party—those must be specified.requires authors to make all data underlying the findings described in their manuscript fully available without restriction, with rare exception, at the time of publication. The data should be provided as part of the manuscript or its supporting information, or deposited to a public repository. For example, in addition to summary statistics, the data points behind means, medians and variance measures should be available. If there are restrictions on publicly sharing data—e.g. participant privacy or use of data from a third party—those must be specified.

Reviewer #1: No

Reviewer #2: Yes

5. Is the manuscript presented in an intelligible fashion and written in standard English?

Reviewer #1: Yes

Reviewer #2: Yes

6. Review Comments to the Author

You may also provide optional suggestions and comments to authors that they might find helpful in planning their study.

Reviewer #1: The NEURAL study is a large, multicenter prospective observational project designed to determine the true incidence of complications—such as nerve injury, hematoma, pneumothorax, and LAST—following single-shot regional anesthesia. It will enroll over 3,396 adult patients across more than 40 Italian centers, with follow-up extending up to one year for persistent neurological deficits. The study collects detailed patient, procedural, and follow‑up data to identify both overall incidence and modifiable risk factors. Ultimately, its findings aim to improve patient safety, refine clinical practices, and support updated evidence‑based guidelines.

Minor revisions:

1. Lack of methods to account for clustering

The protocol does not address the implications of enrolling patients across more than 40 centers, where outcomes may be correlated within institutions due to shared practices, operator skill, equipment, or workflow. Without incorporating statistical methods that account for clustered data—such as multilevel (mixed‑effects) models, generalized estimating equations (GEE), or random‑effects logistic regression—standard errors may be underestimated, leading to inflated Type I error rates. Clarifying how center‑level variation will be handled is essential.

2. No stated strategy for handling missing data

Given the extended follow‑up period (up to one year for neurological deficits), loss to follow‑up and incomplete covariate data are likely. The protocol does not describe any approach for addressing missingness, such as multiple imputation, inverse‑probability weighting, or sensitivity analyses. A predefined plan is necessary to ensure unbiased estimates and transparent reporting.

3. Absence of multiple‑comparison considerations

With numerous block types, predictors, and secondary outcomes, the probability of false‑positive findings increases. Although it is reasonable for exploratory observational studies to avoid overly conservative adjustments, the protocol should acknowledge this issue and clarify whether procedures such as Bonferroni correction, false discovery rate control, or hierarchical testing will be considered in interpreting results.

4. Rare‑event logistic regression concerns

The expected incidence of complications (~0.5%) presents challenges for conventional logistic regression, which can yield biased or unstable estimates when events are sparse. The protocol does not discuss alternative methods tailored to rare events, such as Firth‑penalized likelihood or exact logistic regression, which may improve model performance and inferential reliability.

Reviewer #2: Dear Authors!

Seems to be a well planed study. I don't say it is absolute necessary, but worth thinking to incorporate the early signs of local anaesthetic systemic toxictiy (or is it included in neurological alteration?) and whether they needed intervention or not.

7. PLOS authors have the option to publish the peer review history of their article (what does this mean?). If published, this will include your full peer review and any attached files.). If published, this will include your full peer review and any attached files.

.

Reviewer #1: No

Reviewer #2: **Yes:** Istvan BataiIstvan Batai

---

## [Author Response · Author response to Decision Letter 1]

19 Feb 2026

EDITORIAL DECISION

Q1:This is a well-written and engaging manuscript that addresses an important topic. The study is clearly structured, and the findings are presented in a coherent manner. After careful evaluation of the reviewers’ comments and a thorough assessment of the overall quality, originality, and relevance of the submission, the editorial decision is: MINOR REVISION.

The reviewers have provided constructive suggestions aimed primarily at improving clarity, presentation, and minor methodological or interpretative aspects. Please find attached reviewer’s comments:

A1: Dear Editor, thank you for your kind comments. We have tried to respond to Reviewers comments and we believe the overall quality of the manuscript has improved. We are open to further modifications.

REVIEWER 1 – MINOR REVISION

Q1:The NEURAL study is a large, multicenter prospective observational project designed to determine the true incidence of complications—such as nerve injury, hematoma, pneumothorax, and LAST—following single-shot regional anesthesia. It will enroll over 3,396 adult patients across more than 40 Italian centers, with follow-up extending up to one year for persistent neurological deficits. The study collects detailed patient, procedural, and follow up data to identify both overall incidence and modifiable risk factors. Ultimately, its findings aim to improve patient safety, refine clinical practices, and support updated evidence based guidelines.

A1: We would like to thank the Reviewer for his time and dedication in improving our manuscript

Minor revisions:

Q1. Lack of methods to account for clustering

The protocol does not address the implications of enrolling patients across more than 40 centers, where outcomes may be correlated within institutions due to shared practices, operator skill, equipment, or workflow. Without incorporating statistical methods that account for clustered data—such as multilevel (mixed effects) models, generalized estimating equations (GEE), or random effects logistic regression—standard errors may be underestimated, leading to inflated Type I error rates. Clarifying how center level variation will be handled is essential.

A1: We agree with the comment. We have added the following paragraph:”Accounting for Center-Level Clustering

Given the multicenter design of the NEURAL study, patients are clustered within participating institutions, and outcomes may be correlated due to shared clinical practices, operator expertise, equipment, or local protocols. To account for this hierarchical structure, multilevel (mixed-effects) regression models will be used for the primary and secondary analyses. Specifically, random-intercept logistic regression models will be applied to account for between-center variability when estimating associations between predictors and complications. The intraclass correlation coefficient (ICC) will be calculated to quantify the degree of clustering. As a sensitivity analysis, generalized estimating equations (GEE) with robust standard errors will also be considered.”

Q2. No stated strategy for handling missing data

Given the extended follow up period (up to one year for neurological deficits), loss to follow up and incomplete covariate data are likely. The protocol does not describe any approach for addressing missingness, such as multiple imputation, inverse probability weighting, or sensitivity analyses. A predefined plan is necessary to ensure unbiased estimates and transparent reporting.

A2: Missed data will be handled as shown below:

“If the proportion of missing data exceeds 5% for key variables, multiple imputation by chained equations will be performed under the assumption of missing at random. Imputation models will include all variables used in the primary analyses, including outcomes and relevant predictors. Sensitivity analyses will be conducted to compare complete-case analyses with imputed datasets. In addition, if differential loss to follow-up is observed, inverse probability weighting methods may be applied to assess the robustness of the findings. The amount of missing data and the methods used to address it will be transparently reported according to STROBE recommendations.”

Q3. Absence of multiple comparison considerations

With numerous block types, predictors, and secondary outcomes, the probability of false positive findings increases. Although it is reasonable for exploratory observational studies to avoid overly conservative adjustments, the protocol should acknowledge this issue and clarify whether procedures such as Bonferroni correction, false discovery rate control, or hierarchical testing will be considered in interpreting results.

A3: We agree, added the following to the method section:

“Given the exploratory nature of this large observational study and the evaluation of multiple block types, predictors, and secondary outcomes, we recognize the potential risk of false-positive findings due to multiple comparisons. The secondary outcomes and subgroup analyses, effect sizes and 95% confidence intervals will be emphasized rather than sole reliance on p-values. Where appropriate, false discovery rate control using the Benjamini-Hochberg procedure may be applied to secondary analyses to reduce the risk of Type I error inflation. All secondary analyses will be interpreted cautiously and considered hypothesis-generating.

“Q4. Rare event logistic regression concerns

The expected incidence of complications (~0.5%) presents challenges for conventional logistic regression, which can yield biased or unstable estimates when events are sparse. The protocol does not discuss alternative methods tailored to rare events, such as Firth penalized likelihood or exact logistic regression, which may improve model performance and inferential reliability.

A4: We agree, and we have modified the statistical section that now reads:”

To determine the relationships between the dependent categorical variable (e.g., complications) and one or more independent categorical variables (i.e., complication predictors), logistic regression will be performed to calculate odds ratios (OR) with 95% confidence intervals (CI). Given the anticipated low incidence of complications, rare-event bias may affect conventional maximum likelihood estimation. Therefore, penalized likelihood approaches—such as Firth’s bias-reduced logistic regression—will be used when appropriate to improve the stability and validity of parameter estimates. In cases of extremely sparse data, exact logistic regression methods may also be considered. Model adequacy will be assessed using goodness-of-fit statistics and calibration plots. The presence of multicollinearity will be assessed using variance inflation factors.”

REVIEWER 2 – ACCEPT

Q1:Dear Authors!

Seems to be a well planed study. I don't say it is absolute necessary, but worth thinking to incorporate the early signs of local anaesthetic systemic toxictiy (or is it included in neurological alteration?) and whether they needed intervention or not.

A1: Dear Reviewer, thank you for the time you spent reading our manuscript. Early signs of toxicity are already incorporated in neurological alteration and have been defined as each symptoms that the treating physician recognized as local anesthetic systemic toxicity

---

## [Decision Letter · Decision Letter 1]

18 Apr 2026

Neurological events and unanticipated risks after locoregional anesthesia (NEURAL): protocol for a multicenter prospective observational study

PONE-D-25-62561R1

Dear Dr. De Cassai,

We’re pleased to inform you that your manuscript has been judged scientifically suitable for publication and will be formally accepted for publication once it meets all outstanding technical requirements.

An invoice will be generated when your article is formally accepted. Please note, if your institution has a publishing partnership with PLOS and your article meets the relevant criteria, all or part of your publication costs will be covered. Please make sure your user information is up-to-date by logging into Editorial Manager at Editorial Manager® and clicking the ‘Update My Information' link at the top of the page. For questions related to billing, please contact  and clicking the ‘Update My Information' link at the top of the page. For questions related to billing, please contact billing support..

Kind regards,

Richa Gupta

Academic Editor

PLOS One

Additional Editor Comments (optional):

The authors have adequately addressed all the concerns raised by the reviewers. The revisions have improved the clarity, accuracy, and overall quality of the manuscript. In its current form, the article is suitable for acceptance.

Reviewers' comments:

Reviewer's Responses to Questions

**Comments to the Author**

1. Does the manuscript provide a valid rationale for the proposed study, with clearly identified and justified research questions?

Reviewer #1: Yes

2. Is the protocol technically sound and planned in a manner that will lead to a meaningful outcome and allow testing the stated hypotheses?

Reviewer #1: Yes

3. Is the methodology feasible and described in sufficient detail to allow the work to be replicable?

Reviewer #1: Yes

4. Have the authors described where all data underlying the findings will be made available when the study is complete?

The PLOS Data policy requires authors to make all data underlying the findings described in their manuscript fully available without restriction, with rare exception, at the time of publication. The data should be provided as part of the manuscript or its supporting information, or deposited to a public repository. For example, in addition to summary statistics, the data points behind means, medians and variance measures should be available. If there are restrictions on publicly sharing data—e.g. participant privacy or use of data from a third party—those must be specified.requires authors to make all data underlying the findings described in their manuscript fully available without restriction, with rare exception, at the time of publication. The data should be provided as part of the manuscript or its supporting information, or deposited to a public repository. For example, in addition to summary statistics, the data points behind means, medians and variance measures should be available. If there are restrictions on publicly sharing data—e.g. participant privacy or use of data from a third party—those must be specified.

Reviewer #1: Yes

5. Is the manuscript presented in an intelligible fashion and written in standard English?

Reviewer #1: Yes

7. PLOS authors have the option to publish the peer review history of their article (what does this mean?). If published, this will include your full peer review and any attached files.). If published, this will include your full peer review and any attached files.

.

Reviewer #1: No

---

## [Editor Report · Acceptance letter]

PONE-D-25-62561R1

PLOS One

Dear Dr. De Cassai,

I'm pleased to inform you that your manuscript has been deemed suitable for publication in PLOS One. Congratulations! Your manuscript is now being handed over to our production team.

Kind regards,

on behalf of

Dr. Richa Gupta

Academic Editor

PLOS One